# Microbial Hyaluronic Acid Production: A Review

**DOI:** 10.3390/molecules28052084

**Published:** 2023-02-23

**Authors:** Mónica Serra, Ana Casas, Duarte Toubarro, Ana Novo Barros, José António Teixeira

**Affiliations:** 1Mesosystem, Rua da Igreja Velha 295, 4410-160 Vila Nova de Gaia, Portugal; 2CEB-Centre of Biological Engineering, University of Minho, 4710-057 Braga, Portugal; 3LABBELS–Associate Laboratory, University of Minho, 4710-057 Braga, Portugal; 4CBA and Faculty of Sciences and Technology, University of Azores, Rua Mãe de Deus No 13, 9500-321 Ponta Delgada, Portugal; 5Centre for the Research and Technology of Agro-Environmental and Biological Sciences (CITAB)), Institute for Innovation, Capacity Building and Sustainability of Agri-Food Production (Inov4Agro), University of Trás-os-Montes and Alto Douro (UTAD), Quinta de Prados, 5000-801 Vila Real, Portugal

**Keywords:** hyaluronic acid, fermentation, microbial production

## Abstract

Microbial production of hyaluronic acid (HA) is an area of research that has been gaining attention in recent years due to the increasing demand for this biopolymer for several industrial applications. Hyaluronic acid is a linear, non-sulfated glycosaminoglycan that is widely distributed in nature and is mainly composed of repeating units of N-acetylglucosamine and glucuronic acid. It has a wide and unique range of properties such as viscoelasticity, lubrication, and hydration, which makes it an attractive material for several industrial applications such as cosmetics, pharmaceuticals, and medical devices. This review presents and discusses the available fermentation strategies to produce hyaluronic acid.

## 1. Introduction

Hyaluronic acid (HA) is a naturally occurring biopolymer that is widely distributed in nature. It is a linear, non-sulfated glycosaminoglycan that is composed mainly of repeating units of N-acetylglucosamine and glucuronic acid linked by β- (1-4) and β- (1-3) glycosidic bonds making its structure energetically stable (Figure 1) [1]. Each disaccharide presents a molecular weight (MW) of around 400 Da, and a hyaluronic acid chain can be composed of 10,000 disaccharides, which means a molecular weight of around 4.0 × 10^3^ kDa [2,3].

Hyaluronic acid plays an important role in living organisms and is an attractive biomaterial for different applications due to its features, in particular moisturizing retention ability, viscoelasticity, resistance to mechanical damage, and lack of immunogenicity and toxicity. In many cases, it acts as a lubricant (joints), a structure stabilizer, an organ space filler (skin), and a shock absorber (cartilage) [4,5,6].

It is a naturally occurring polysaccharide found in the body that plays a vital role in skin health and beauty. It is a major component of the extracellular matrix (ECM) and is responsible for maintaining skin hydration, elasticity, and volume [5,7]. As we age, the levels of hyaluronic acid in our skin decrease, leading to wrinkles, dryness, and loss of firmness. Therefore, the use of products containing hyaluronic acid has become increasingly popular in the cosmetic industry to combat the signs of ageing and promote healthy, youthful-looking skin [8].

This biocompatible polymer immobilizes the water in the tissue, and it can change the dermal volume that influences cell proliferation, differentiation, and tissue repair. The biological functions depend on its molecular weight (MW), for example, mucoadherence is a property of hyaluronic acid with a high molecular weight. This type of hyaluronic acid is used as space fillers, antiangiogenic and immunosuppressive, while medium-size hyaluronic acid chains are involved in ovulation, embryogenesis, and wound repair. Small chains of hyaluronic acid have inflammatory, immuno-stimulatory, angiogenic, and anti-apoptotic properties [3,7,9].

Hyaluronic acid is a humectant, which means it attracts and retains moisture. It can hold up to 1000 times its weight in water, making it an effective ingredient in moisturizers and serums. By keeping the skin hydrated, hyaluronic acid helps to plump up the skin and reduce the appearance of fine lines and wrinkles. It also improves skin elasticity, making it appear firmer and more youthful. In addition to its ability to hydrate and plump the skin, hyaluronic acid has anti-inflammatory properties that can help to reduce redness and irritation. This makes it a great ingredient for sensitive skin types, as well as for those with conditions such as rosacea or eczema [8,10].

The applications of hyaluronic acid are diverse such as the correction of facial folds and wrinkles, body contouring, and as a marker in the diagnosis of tumours. Hyaluronic acid can also be used in the supplementation of joint fluid, in eye surgery, regeneration of surgical wounds, and as a drug delivery agent for various administration routes [2,4,11,12].

Hyaluronic acid can also be used to enhance the effects of other skincare ingredients. For example, when combined with retinoids, it can help to reduce the dryness and irritation that can be caused by these powerful anti-ageing ingredients [13]. Similarly, when used in combination with antioxidants, it can help to protect the skin from environmental damage [10,14].

While hyaluronic acid is a naturally occurring substance in the body, topical products containing hyaluronic acid are typically derived from either rooster combs or fermentation. There are different types of hyaluronic acid, such as sodium hyaluronate and hyaluronic acid. Sodium hyaluronate is a salt form of hyaluronic acid that is smaller and can penetrate deeper into the skin [15].

The demand and the value of hyaluronic acid have increased over the years. According to R. G. Ferreira et al. [16], the estimated market of hyaluronic acid was EUR 7.6 billion in 2019. It is expected to have an annual growth of 8.1% from 2016 to 2027, which means that 20 MT of hyaluronic acid produced by that year will have an average price between EUR 1500/kg and EUR 4000/kg. The main market segments are derma fillers in cosmetics, osteoarthritis, and ophthalmology. Consequently, it is necessary to understand the improvement points in the hyaluronic acid production process, including downstream processing [16].

## 2. Hyaluronic Acid Production Using the Fermentation Process

Microbial production of hyaluronic acid is an area of research that has been gaining attention in recent years. Hyaluronic acid is widely distributed in nature [1], and it has a wide range of properties such as viscoelasticity, lubrication, and hydration that make it an attractive material for various industrial applications [2,4,11,12].

Microorganisms such as bacteria and yeast have been used to produce HA through fermentation. *Streptococcus zooepidemicus*, a Gram-positive bacteria, is one of the most widely used organisms for the production of HA due to its high hyaluronic acid production rate and ease of cultivation [5,7,17].

Traditionally, hyaluronic acid is extracted from animal sources such as rooster combs and cocks’ combs. However, the difficulty in controlling animal tissue, high costs, and ethical concerns associated with animal-derived HA have led to the development of microbial production methods [5,7,9].

During the production of hyaluronic acid, there are still challenges to be overcome, such as the limited production of hyaluronic acid due to the high viscosity of the broth, causing difficulties in the mixing and mass transfer rate of oxygen; competition for the same precursors for cell growth and hyaluronic acid production; and the accumulation of lactic acid, the main by-product of HA fermentation, causing the inhibition of cell growth and hyaluronic acid production [5,7,9].

To overcome these challenges and also increase the microbial production of hyaluronic acid, several strategies have been studied: selection of producing strains and appropriate culture media, the establishment of culture methods, and determination of culture conditions, among others.

## 3. Selection of a Microorganism Producer and Its Cultivation Media

The selection of the appropriate microorganism to produce hyaluronic acid is an important factor in the microbial production process. Each microorganism has its own unique advantages and disadvantages in terms of production performance and profitability. When choosing a microorganism to produce hyaluronic acid, it is essential to consider the specific requirements of the final application of this molecule [5,7,9].

The microorganism must be able to produce high yields of hyaluronic acid, be easy to cultivate, and have a low cost of production. Several microorganisms have been used to produce hyaluronic acid through fermentation, including *Streptococcus zooepidemicus*, *Bacillus subtilis*, and *Escherichia coli* [18,19,20,21].

*Streptococcus zooepidemicus* produces a high molecular weight HA with excellent biocompatibility, making it suitable for medical applications such as wound healing and joint lubrication. However, *Streptococcus* is a slow-growing organism, and the production process is costly due to the need for expensive media and downstream processing. However, it does not require the use of toxic chemicals or solvents, resulting in a pure and safe end product [16,22].

*Bacillus subtilis* is another microorganism that has been used to produce hyaluronic acid. It is a Gram-positive bacterium that is known for its ability to produce high yields of HA and is also non-pathogenic. However, it is considered more difficult to cultivate than *Streptococcus zooepidemicus. Bacillus* produces a lower molecular weight hyaluronic acid that is suitable for cosmetic applications such as moisturizing creams and serums. *Bacillus* grows rapidly and is relatively cheap to produce, making it a more profitable option for cosmetic companies [18,23].

*Escherichia coli* is a Gram-negative bacterium that has been used for the production of hyaluronic acid. It is a well-studied organism and has a wide range of genetic tools available for use in genetic engineering. However, it produces a lower yield of hyaluronic acid compared to *Streptococcus zooepidemicus* and *Bacillus subtilis* [20,21,24].

Another microorganism that has been recently explored for hyaluronic acid production is the yeast *Pichia pastoris*. This organism produces a high yield of low molecular weight HA, making it suitable for use in the pharmaceutical industry. The production process is relatively cheap and can be scaled up easily, making it a promising candidate for large-scale production [25,26].

When considering the profitability and possible applications of the hyaluronic acid produced by each microorganism, it is essential to consider factors such as production costs, purity of the final product, and market demand. For example, the high-molecular-weight hyaluronic acid produced by *Streptococcus* has significant potential in the medical field due to its excellent biocompatibility, making it a high-value product. Meanwhile, the lower-molecular-weight hyaluronic acid produced by *Bacillus* has excellent potential in the cosmetic industry, with demand for hyaluronic acid-based cosmetics continuing to grow.

In conclusion, the selection of the appropriate microorganism for hyaluronic acid production is critical for determining the production performance and profitability of the process. By evaluating the advantages of the final application, producers can optimize their production process, enhance product quality, and increase profitability.

Group A and C *streptococci*, namely *Streptococcus zooepidemicus*, have been the most explored strain in the production process of microbial hyaluronic acid and have obtained the best results. Nevertheless, due to the pathogenicity of this natural HA-producer bacteria, other metabolically engineered microorganisms have been studied, such as *Bacillus subtilis* [18,19,23], *Corynebacterium glutamicum* [27,28,29], *Escherichia Coli* [20,21,24], *Lactococcus lactis* [30,31], *Pichia pastoris* [25], and *Kluyveromyces lactis* [32]. Among the cited strains, *Corynebacterium glutamicum* has presented the best results [28,29].

The cultivation medium to produce hyaluronic acid A is also an important factor to consider. The medium must contain a source of carbon, nitrogen, vitamins, minerals, and other growth factors for the microorganism. Additionally, the medium should be optimized to promote the production of hyaluronic acid by the microorganism [5,26].

Concerning media composition for hyaluronic acid production, an important factor is the cultivation condition. *Streptococcus zooepidemicus* is a nutritionally demanding microorganism, and a nitrogen source is an essential nutrient for its growth. This bacterium does not synthesize some amino acids that are favourable for its growth and hyaluronic acid production. Therefore, media supplementation with nutrients, such as amino acids, has been studied [5,26].

### 3.1. Alternative Sources as Substrates for the Culture Media

One of the key factors in the production of hyaluronic acid is the culture medium that is used to grow the microorganism producers.

Traditionally, the culture medium for hyaluronic acid production has been based on complex sugars such as glucose and fructose, but these can be expensive and may not be readily available [33,34]. As a result, researchers have been investigating alternative sources as substrates for the culture medium. Some of the most promising alternative sources include:

Agricultural waste: By-products of agriculture, such as sugar beet pulp, corn steep liquor, and wheat bran, have been found to be suitable substrates for hyaluronic acid production. These waste products are readily available and have the added benefit of being environmentally friendly [35].Industrial waste: Industrial waste products, such as distillery waste and molasses, have also been found to be suitable substrates for hyaluronic acid production. These waste products are often cheaper than traditional substrates, making them an attractive option for commercial production [35,36].Synthetic substrates: Some researchers have also investigated synthetic substrates, such as hydrolysates of starch and cellulose, as an alternative substrate for hyaluronic acid production. These substrates are relatively cheap and easy to produce, making them an attractive option for commercial production [37].

Overall, alternative sources as substrates for the culture media for hyaluronic acid production have been explored to reduce the costs of production and make the process more sustainable. While more research is needed to fully understand the potential of these alternative sources, they offer promising possibilities for the future of hyaluronic acid production.

Different agricultural resources and industrial wastes have been explored as alternative nutritive sources for microbial hyaluronic acid. The goal of the formulation of cost-effective culture media is to maintain low costs during microbial hyaluronic acid production and to reduce pollution problems, as well as to improve the efficiency of the fermentation processes.

Amado et al. [36] studied the optimization of a media containing cheese whey to produce hyaluronic acid by *Streptococcus zooepidemicus*. The major nutrients in cheese whey are lactose, soluble proteins (β-lactoglobulin, α-lactalbumin), lipids, and B-group vitamins. Using cheese whey protein and glucose as nitrogen and carbon sources, respectively, the maximum production rate was 0.87 g/L h (0.75 g/L h in control media with glucose and yeast extract), the HA production was 4.02 g/L (3.19 g/L in control media), and the HA average molecular weights (HA-MW) were 3.71 × 10^3^ kDa.

In another study, Amado et al. [35] formulated a culture media containing corn steep liquor (CSL) instead of tryptone as a nitrogen source. The highest hyaluronic acid production in this media was 3,48 g/L (comparable to the control media of 3.60 g/L) with a molecular weight of 3.8 × 10^3^ kDa (higher than the control media of 3.0 × 10^3^ kDa).

Arslan & Aydogan [38] tested the effectiveness of sheep wool peptone (SWP) and molasses as nitrogen and carbon sources, respectively, in fermentation media to produce hyaluronic acid, using the same strain as Amado et al. They verified a higher HA production in media containing sheep wool peptone (3.54 g/L) than in media containing tryptone TP (2.58 g/L) and peptone PP (2.47 g/L). Sheep wool peptone has a lower protein content (70.6 g/100 g) than tryptone and peptone (83.1 and 83.3 g/100 g, respectively), and by contrast, sheep wool peptone has higher element contents (K, P, and Mg) than tryptone and peptone, and these elements promote hyaluronic acid production. Moreover, this nitrogen source contains high cystine and arginine contents, which are the main amino acids that affect hyaluronic acid production. These are possible reasons for the higher production of hyaluronic acid in sheep wool peptone than in tryptone and peptone.

Similarly, Pan et al. [39] developed a media for hyaluronic acid production which contains sugarcane molasses. After molasses treatment with activated charcoal and using it as a substrate for *Streptococcus zooepidemicus*, the hyaluronic acid production was 19% higher than using molasses without treatment. This suggested pre-treatment decreased the excessive metal ions content that can inhibit hyaluronic acid production. After 24 h of fermentation with pH control (pH 8), the hyaluronic acid production was 2.83 g/L.

Pires et al. [40] investigated the use of cashew apple juice in the fermentation media, and the HA production was 0.89 g/L with a molecular weight of 18.4 kDa.

Vázquez et al. [41] explored a culture media using marine by-products as substrates for *Streptococcus zooepidemicus* when producing microbial hyaluronic acid. Firstly, glycogen from mussel processing wastewater (MPW) and tuna peptone from viscera residue were used as a carbon source and a protein source, respectively. In this media, hyaluronic acid production and the cell growth rate were lower (2.46 g/L and 0.81 g/L/h, respectively) than in commercial media containing glucose and tryptone (3.07 g/L and 1.32 g/L/h, respectively); on the other hand, the hyaluronic acid produced in alternative media presented a higher molecular weight (2.50 × 10^3^ kDa). Later, the same scientists explored a media formulated with peptones obtained from *Scyliorhinus canicula* viscera by-products. Using this alternative substrate as a nitrogen source, hyaluronic acid production was explored in fed-batch culture. Using this culture mode, the hyaluronic acid production (2.53 g/L) and molecular weight (2.11 × 10^3^ kDa) of the polysaccharide were increased compared to production using the batch mode (2.26 g/L; 1.80 × 10^3^ kDa). Compared with the commercial media (3.23 g/L; 0.930 g/L/h; 1.85 × 10^3^ kDa), the hyaluronic acid production and cell growth rate were lower in the alternative media; however, the hyaluronic acid molecular weight was higher [42].

Benedini & Santana [43] studied the effect of replacing Brain Heart Infusion (BHI) with soy peptone (SP) as a nitrogen source for *Streptococcus zooepidemicus*. In this case, there was an increment in hyaluronic acid production and hyaluronic acid molecular weight from 0.29 g/L and 3.09 × 10^3^ kDa to 0.30 g/L and 3.60 × 10^3^ kDa, respectively. The increase in hyaluronic acid molecular weight was related to the diminution of lactic acid production and hyaluronic acid exposition to a higher pH.

Another study was performed in which Ghodke et al. [44] evaluated palmyra palm sugar (Pj) and soya peptone to formulate a media for hyaluronic acid production by *Streptococcus zooepidemicus*. Palmyra palm sugar contains sucrose and vitamins (nicotinic acid, thiamine, riboflavin, and vitamin C) that are essential to the growth of a microorganism, which led to the hyaluronic acid production and specific growth rate on palmyra palm sugar-based media (0.41 g/L; 0.54 h^−1^) being higher than on pure sucrose-based media (0.31 g/L; 0.32 h^−1^). After exploring the best initial substrate concentration (Pj, 30 g/L), the microorganism yielded a hyaluronic acid concentration of 1.22 g/L with 9.50 × 10^2^ kDa.

Zhang et al. [37] proposed a serum-free starch media in which the cells could grow and produce hyaluronic acid. Replacing glucose with starch obtained hyaluronic acid with a yield of 6.7 g/L. The studied strain could grow in serum-free media but not in other carbohydrates such as glucose. This could be due to the reduction in lactic acid (metabolite from glucose catabolism) in the broth for decreasing glucose in the media.

In their work, Duffeck et al. [45] optimized the production of hyaluronic acid in sugarcane molasses pre-treated with active charcoal media (likewise explored by Pan et al. [39]) with nutrient supplementation (glutamine). They concluded that the hyaluronic acid production by *Streptococcus zooepidemicus* was higher in sugarcane molasses media (0.710 g/L) than in glucose media (0.469 g/L) due to the molasses media being rich in sugar and amino acids, and after treatment, this molasses contains fewer inhibitors to hyaluronic acid production. Regarding media supplementation, glutamine supplementation revealed positive effects on hyaluronic acid production, improving the yield of the product to 2.55 g/L, because this amino acid acts as the amino donor group to UDP-N-acetylglucosamine formation (one of hyaluronic acid precursors).

### 3.2. Supplementation of Culture Media

The process of media supplementation in the production of hyaluronic acid includes:Carbon source supplements: Carbon sources such as glucose, fructose, and maltose can be added to the culture medium to provide an additional energy source for the microorganisms. This can help increase the rate of hyaluronic acid production [46].Nitrogen source supplements: Nitrogen sources such as ammonium sulfate and yeast extract can be added to the culture medium to provide essential amino acids and other nutrients for the microorganisms. This can help increase the rate of hyaluronic acid production [33].Vitamin supplements: Vitamins such as thiamine, riboflavin, and pyridoxine can be added to the culture medium to promote the growth of microorganisms and increase the rate of hyaluronic acid production [47].Mineral supplements: Minerals such as sodium, potassium, and calcium can be added to the culture medium to provide essential minerals for the microorganisms and promote hyaluronic acid production [34].Growth factors: Growth factors such as yeast extract, soy peptone, and tryptone can be added to the culture medium to provide additional nutrients and growth factors for the microorganisms [33]. This can help increase the rate of hyaluronic acid production.

It is important to note that the appropriate supplements and the optimal concentration of the supplement in the culture medium can vary depending on the microorganism used for hyaluronic acid production. Researchers often use a combination of different supplements and tweak the concentration to find the best condition to produce hyaluronic acid. It is also important to note that the use of supplements in the culture medium can increase production costs; hence, it is important to find a balance between costs and the enhancement of production.

The substrates used more often in hyaluronic acid production are glucose and sucrose as the primary carbon and energy sources and yeast extract and peptones as nitrogen sources [33,34,48,49,50,51,52]. Other carbon sources have been explored as substrates for hyaluronic acid production, namely, lactose, maltose, galactose, mannose, and others; however, hyaluronic acid production is not as high [18,34,53,54,55].

In their investigation, Chen et al. [56] used glucose as a carbon source and yeast extract as a nitrogen source to explore the best carbon-to-nitrogen (C/N) ratio and concluded that the C/N ratio of 2:1 maximizes hyaluronic acid synthesis at 2.45 g/L. In other C/N ratios, such as 1,3:1 and 4:1, the hyaluronic acid production was 2.20 g/L and 1.52 g/L, respectively.

Concerning media supplementation, Aroskar et al. [46] also investigated the effect of different nutrients in the culture media for *Streptococcus zooepidemicus* production of hyaluronic acid. In this study, the carbon source used was dextrose for producing hyaluronic acid with a higher yield (0.70 g/L) than other carbon sources used, such as sucrose (0.51 g/L), maltose (0.50 g/L), and dextrin (0.55 g/L). The researchers explored the addition of L-arginine HCl and tannic acid to the fermentation media, which resulted in an increment in the yield of HA to 1.029 g/L. L-arginine HCl is a carbon and nitrogen donor for the synthesis of nucleotides that are needed for the growth and multiplication of microorganisms, thus saving energy consumption in the organism for its production. Tannic acid represses the synthesis of the hyaluronidase enzyme that acts by depolymerizing hyaluronic acid. In light of this, the use of tannic acid showed an increase in the molecular weight of hyaluronic acid.

In the same way, Shah et al. [33] explored the supplementation media for *Streptococcus zooepidemicus* with glutamine and sodium iodoacetate. Iodoacetate decreases lactic acid synthesis by inhibiting the glycolysis pathway; consequently, there was a redirection of the carbon flow from lactic acid formation to UDP glucuronic acid formation, one of the precursors of hyaluronic acid. Glutamine, as mentioned above, is an amino acid involved in the formation of one of the hyaluronic acid precursors. Accordingly, the HA concentration was increased from 2 g/L in media without supplementation to 5 g/L, the specific growth rate was decreased from 0.42 h^−1^ to 0.25 h^−1^, and the hyaluronic acid molecular weight was increased from 2.4 × 10^3^ kDa to 3.2 × 10^3^ kDa.

Aiming to obtain HA with high yield and molecular weight, Im et al. [34] studied the optimization of media components using the strain *Streptococcus* sp. ID9102 (KCTC 11935BP), which is negative to hemolytic activity and hyaluronidase. Exploring the best carbon source, a higher hyaluronic acid production was reached when glucose was the carbon source (1.58 g/L) followed by lactose and sucrose, which reached similar results (≈1.3 g/L). According to the results, the investigators selected the best components media as glucose, yeast extract, casein peptone, K_2_HPO_4_, and MgCl_2_ and obtained a hyaluronic acid production of 5.88 g/L with a molecular weight of 3.60 × 10^3^ kDa. After supplementing the media with amino acids (glutamine and glutamate) and organic acid (oxalic acid), the HA titer and HA-MW were higher (6.94 g/L and 5.90 × 10^3^ kDa, respectively).

Kim et al. [47] also studied the production of HA from *Streptococcus equi* mutant KFCC 10830. This mutant has nonhemolytic and hyaluronidase-negative characteristics. Furthermore, the scientists explored the effect of lysozyme addition during cultivation on the fermentation media. Lysozyme can damage the cell walls of hyaluronic acid producer microorganisms, which then produce more hyaluronic acid to protect themselves from the action of lysozyme. According to this, the addition of lysozyme increased the production and molecular weight of hyaluronic acid.

## 4. Determination of Culture Conditions

Another important factor to develop a successful process for producing microbial hyaluronic acid is identifying ideal fermentation parameters.

The culture conditions refer to the various factors that can affect the growth and production of the microorganisms used in hyaluronic acid production. Some of the key culture conditions that must be determined include:Temperature: The optimal temperature for the growth of the microorganisms and the production of hyaluronic acid will vary depending on the species used. The temperature range that is most suitable to produce HA is usually between 30 and 37 °C [50].pH: The pH of the culture medium is another important culture condition that must be determined. The pH must be maintained within a specific range that is suitable for the growth of the microorganisms and the production of hyaluronic acid. The optimal pH range for HA production is usually between 6.5 and 7.5 [57,58].Aeration: Adequate aeration is necessary for the growth of the microorganisms and the production of hyaluronic acid. The amount of air supplied to the culture medium can be adjusted to achieve the optimal aeration for the microorganisms used [59].Agitation: Agitation is the process of physically stirring the culture medium to ensure that the microorganisms are well-mixed and have access to nutrients and oxygen. The optimal agitation rate will depend on the microorganisms used and the culture medium [49].Substrate concentration: The concentration of the substrate in the culture medium will affect the rate of hyaluronic acid production. The optimal substrate concentration will depend on the microorganisms used and the culture medium [51].Supplementation: Supplementation of the culture medium with various nutrients and growth factors can also affect the rate of hyaluronic acid production. The optimal concentration of supplements will depend on the microorganisms used and the culture medium [48,49,60].

Determining the optimal culture conditions to produce hyaluronic acid will typically involve a process of experimentation and optimization. Researchers will often use a combination of different culture conditions and tweak the various factors to find the conditions that yield the highest hyaluronic acid production [57,58,59,60,61].

In their work, Johns et al. [57] identified the optimum pH as 6.7. A high agitation rate of 600 rpm was found to be advantageous to release the HA capsule from *Streptococcus zooepidemicus* bacteria cells into the medium; however, a higher shear rate can damage the polymer. In addition, the use of aerated cultures enhanced the hyaluronic acid concentration, maybe due to the highest energy yield.

David C. Armstrong & Johns [50] described the effect of different culture conditions on the molecular weight of HA produced by *Streptococcus zooepidemicus*. Concerning temperature, their results showed that lower temperatures (32 °C) resulted in higher molecular weight and higher yield of the produced hyaluronic acid than higher temperatures (at 40 °C: 1.85 × 10^3^ kDa and 1.20 g/L, respectively), and by decreasing the temperature, the specific growth rate also decreased, as the two processes (cell growth and HA production) are competitors for the same precursors. Their results also showed that higher aeration rates (1.0 vvm) enhanced HA production and hyaluronic acid molecular weight from 1.45 g/L and 2.10 × 10^3^ kDa (0.2 vvm) to 4.20 g/L and 3.0 × 10^3^ kDa, respectively. From the other culture conditions explored, the initial glucose concentration also presented an influence on hyaluronic acid production. A higher initial glucose concentration (60 g/L) in the fermentation broth enhanced the HA concentration and HA-MW, which could be due to HA precursors (UDP-GlcUA and UDP-GlcNAc) being derived from glucose. However, the initial glucose concentration is limited to 60 g/L because at higher concentrations, there are limitations in mass transfer due to the high broth viscosity.

Similarly, Don & Shoparwe [51] explored the effect of glucose in the fermentation media for *S. zooepidemicus* on HA production. The highest hyaluronic acid molecular weight (2.52 × 10^3^ kDa) was reached when the initial glucose concentration was 40 g/L. Above these concentration levels, there was a reduction in hyaluronic acid molecular weight.

To achieve the highest concentration of hyaluronic acid (5.3 g/L), Zakeri & Rasaee [52] identified the best fermentation conditions. This way, after they explored various fermentation conditions, they proposed HA production with the following conditions: 37 °C, pH 7, 300 rpm, and DO (dissolved oxygen) 50%. Moreover, they also identified the best sources of carbon and nitrogen: glucose (70 g/L) and yeast extract (30 g/L).

Using the same substrates as Zakeri & Rasaee [52], Huang et al. [59] explored the role of dissolved oxygen and the effect of agitation on hyaluronic acid production using fermentation. They found an increment in hyaluronic acid yield (from 2.7 g/L to 3.1 g/L) when dissolved oxygen increased from 2.5% to 5%, and above these values of dissolved oxygen, the HA yield was not influenced. So, the investigators found a critical level of dissolved oxygen to produce HA, and they suggested dissolved oxygen to have a stimulant role in hyaluronic acid synthesis. Concerning agitation, its main function is to mix the broth; however, agitation can favour oxygen absorption by cells. At 400 rpm (0.74 g HA/g cell), the hyaluronic acid productivity was higher than at 200 rpm (0.65 g HA/g cell).

Under aerobic conditions at 37 °C and with 200 rpm agitation, L Liu et al. [58] explored an intermittent alkaline-stress strategy. Under stress conditions at pH 8.5 for 1 h to 6 h of fermentation, a hyaluronic acid concentration of 6.5 g/L was obtained (higher than control conditions: 5.0 g/L). This increase in productivity may be due to the redirection of the carbon flow (more availability of carbon source to produce hyaluronic acid rather than cell growth or lactic acid production) and/or the increase in dissolved oxygen that was found in stress conditions.

Oxygen mass transfer has an important role in hyaluronic acid production. Nonetheless, the high viscosity of the media caused by hyaluronic acid production can challenge ideal mixing and uniform oxygenation. To overcome this problem, Saharkhiz & Babaeipour [60] studied the effect of diluting the culture media controlling pH with a low concentration of NaOH. The dilution improved the mass transfer of oxygen, and the hyaluronic acid production was enhanced from 6.6 g/L to 8.4 g/L.

Long Liu et al. [49] proposed three strategies to control dissolved oxygen in the production media. Firstly, they applied a three-stage agitation speed control culture model in which they varied the agitation speed between 200 rpm (0–8 h), 400 rpm (8–12 h), and 600 rpm (12–20 h). The HA productivity was enhanced from 5.0 g/L at constant agitation speed (200 rpm) to 5.5 g/L. This may be due to the increase in the dissolved oxygen level that is favourable to the growth of the cells, and the increase in agitation that induces shear stress to cells. Following the previous model, the investigators explored a two-stage dissolved oxygen control model. In this model, they controlled the dissolved oxygen level at the 10% level during the first 8 h of the fermentation process, and after, they decreased the dissolved oxygen level to 5% using agitation speed control. Once there was a difference in the critical dissolved oxygen levels for cell growth and hyaluronic acid production, with this strategy, it was possible to redirect the carbon flow in the first phase of the fermentation process for cell growth and, later, for the production of hyaluronic acid. As a result, the HA production and productivity increased to 6.3 g/L and 0.0984 g/L/h (from 5 g/L and 0.0694 g/L/h, respectively). Another strategy to improve dissolved oxygen levels in the broth is media supplementation with oxygen vectors. In this way, Long Liu et al. [36] explored the effect of PFC (perfluorodecalin) addition to the media at 8 h of the process. It was found that perfluorodecalin improved the dissolved oxygen levels from 0.5% to 5% at a low agitation speed (200 rpm), and the HA titer in the media was 6.6 g/L.

Likewise, Z. W. Lai et al. [48] investigated the effect of n-hexadecane as an oxygen vector. This oxygen vector enhanced the hyaluronic acid concentration from 2.45 g/L to 4.25 g/L and the hyaluronic acid molecular weight from 5.20 × 10^3^ kDa to 1.54 × 10^4^ kDa. So, this suggested n-hexadecane can be used as an oxygen vector and organic phase for increasing the molecular weight of hyaluronic acid. This investigation of Lai et al. [48] confirmed what was reported in the Duan et al. [62] studies. In their work, they explored the effect of dissolved oxygen levels on the molecular weight of hyaluronic acid and concluded that molecular weight is dependent on the balance of the effect of ATP and reactive oxygen species (ROS) in the culture media. Moreover, higher dissolved oxygen levels enhanced ATP levels and ROS in the media. As it is known, ATP and ROS levels have contrary effects, so the highest HA-MW (2.19 × 10^3^ kDa) was reached on the balance of these two effects at the 50% dissolved oxygen level. Above this level (80% DO), the HA-MW was decreased (2.06 × 10^3^ kDa).

To achieve a balance between a higher hyaluronic acid concentration and a higher hyaluronic acid molecular weight, J. Liu et al. [63] developed a two-stage fermentation strategy. The highest hyaluronic acid concentration and hyaluronic acid molecular weight were reached at the same conditions of aeration rate and agitation speed (1 vvm and 600 rpm, respectively). However, these values of concentration and molecular weight were achieved at different conditions of temperature and pH. The first stage of fermentation was performed at pH 8 and 31 °C allowing for the best conditions for molecular weight growth, and lastly, the final stage of the process was conducted at pH 7 and 37 °C to promote HA accumulation. The HA titer and hyaluronic acid molecular weight increased from 2.99 g/L and 2.26 × 10^3^ kDa to 4.75 g/L and 2.36 × 10^3^ kDa, respectively.

Li et al. [64] evaluated the production of hyaluronic acid with different MW from a single-producing bacterium, *Bacillus subtilis*. The scientists obtained high-, medium-, and low-MW HAs by adjusting the temperature (32 °C, 42 °C, and 47 °C, respectively); the highest molecular weight of hyaluronic acid (6.19 × 10^3^ kDa) was obtained at 47 °C with a hyaluronic acid concentration of 1.88 g/L. However, the highest hyaluronic acid concentration (4.25 g/L) with 8.61 kDa was achieved at 32 °C.

## 5. Fermenter Configuration

Fermenter configuration refers to the design and setup of the equipment used in the fermentation process for the production of hyaluronic acid [65]. A typical fermenter setup includes the following components:The fermenter vessel: This is the main container where the fermentation process takes place. The vessel is typically made of stainless steel and can be designed in a variety of shapes and sizes depending on the scale of the production.Agitator: This is a mechanical device that is used to mix the culture medium and provide oxygen to the microorganisms. Agitators can be designed in a variety of forms, such as propellers, impellers, and turbines, and can be adjusted to provide the optimal agitation rate for the microorganisms used.Aeration system: This is a device that is used to supply oxygen to the culture medium. Aeration systems can be designed in a variety of forms such as spargers, air bubblers, and air diffusers.pH and temperature control: These devices are used to monitor and maintain the pH and temperature of the culture medium within the optimal range for the growth and production of hyaluronic acid.Control system: This is a device that is used to control and monitor the various parameters of the fermentation process such as temperature, pH, agitation rate, and aeration rate.

The design and configuration of the fermenter will depend on the scale of production and the microorganisms used. Additionally, a large-scale industrial fermentation system is equipped with additional features such as automatic sampling, automatic pH and temperature control, automatic foaming control, automatic sterilization, etc.

Several scientists have investigated agitation speed and impeller type to improve the oxygen transfer coefficient. Z. W. Lai et al. [48] explored the effect of the helical ribbon and the Rushton turbine impeller, and they verified that K_L_α (gas–liquid volumetric oxygen transfer coefficient) was enhanced by 25% using the helical ribbon as compared to the Rushton turbine impeller. A decrease in K_L_α is related to the increase in viscosity, which affects the oxygen transfer rate. Using the helical ribbon impeller, the gas–liquid volumetric oxygen transfer coefficient was improved to 1.047 m/s impeller tip speed; however, the fermentation performance improved at a 0.785 m/s impeller tip speed. This decrease in fermentation performance could be due to the high shear rate on bacterial cells.

In their work, Kim et al. [47] tested two different types of impellers: Rushton type and intermig type. The hyaluronic acid production was similar using the two impellers under the same fermentation conditions, but there was an increase in molecular weight of hyaluronic acid using the intermig type (4.8 × 10^3^ kDa) instead of the Rushton type (3.8 × 10^3^ kDa). Despite that, the Rushton-type impeller has been the most used system to agitate fermentation [49,58,66].

## 6. Establishment of Culture Modes

Establishing the modes of culture is an important step in the production of hyaluronic acid using microorganisms. The mode of culture refers to the method used to grow and maintain the microorganisms used in the fermentation process. Some common modes of culture include:Batch culture: In batch culture, the microorganisms are grown in a closed vessel for a specific period of time, after which the culture is harvested, and the hyaluronic acid is extracted. Batch culture is simple and easy to set up, but it has the disadvantage of being less efficient and more costly than other modes of culture [31,56].Fed-batch culture: In fed-batch culture, the microorganisms are grown in a closed vessel and are periodically fed with a specific substrate or supplement. This allows for a higher concentration of microorganisms and a higher rate of hyaluronic acid production. Fed-batch culture is more efficient than batch culture, but it is more complex to set up and control [67].Continuous culture: In continuous culture, the microorganisms are grown in a closed vessel and are continuously supplied with a fresh medium. This allows for a high concentration of microorganisms and a high rate of hyaluronic acid production. Continuous culture is the most efficient mode of culture, but it is also the most complex to set up and control [56].

Each mode of culture has its own advantages and disadvantages, and the most appropriate mode of culture will depend on the microorganisms used and the scale of production. Researchers often use a combination of different modes of culture and tweak several factors to find the conditions that yield the highest hyaluronic acid production.

Fermentation process development includes the establishment of a culture mode. The most used culture mode to produce hyaluronic acid has been the batch mode [31,50,56,67]. In their work, Long Liu et al. [67] explored the effect of different culture modes on hyaluronic acid production, and they found higher hyaluronic acid concentration (5,0 g/L) using the batch mode than using the fed-batch mode (4.72 g/L); however, the cell concentration increased from 13.3 g/L in the batch mode to 14.7 g/L in the fed-batch mode.

These results suggest that the fed-batch mode is preferred for cell growth, and the batch mode is more suitable for producing hyaluronic acid. Consequently, investigators explored a two-stage cultivation strategy in which the cultivation was performed in the fed-batch mode in the early phase of the fermentation process and lastly in batch mode. As a result, the hyaluronic acid concentration was higher (6.6 g/L) than the process performed in batch (5 g/L) and fed-batch culture (4 g/L) [68].

However, the batch mode has some limitations such as the long turnaround time and decreasing hyaluronic acid production due to inhibition by a high carbon source concentration. Thus, Chen et al. [56] proposed a fill-and-draw operation to improve the batch mode culture, such as reducing the turnover time of the fermenter. The investigators found that the operation had better results when performed in the late exponential growth phase (8 h), and the best quantity of displaced media was one-third of the broth. The hyaluronic acid concentration was 1.60 g/L.

Huang et al. [69] suggested another mode to skip the turnaround time on the fermenter and the lag phase of the process: the repeated batch mode. Seeding with 5% of the broth, the cell and hyaluronic acid concentration (2.3 g/L) were maintained as in the batch culture for seven repeated cycles. When the seed was increased, the hyaluronic acid production decreased, which suggested there were inhibitors for hyaluronic acid production in the broth.

To maintain the environmental and intracellular conditions in the fermentation media, Badle et al. [70] explored the chemostat mode. In this mode, they found an inverse relationship between the molecular weight of hyaluronic acid and the dilution rate of the media, and the best dilution rate found was 0.066 h^−1^ to produce HA with 2.6 × 10^3^ kDa. Other dilution rates (0.1, 0.2, and 0.3 h^−1^) lowered the molecular weight of the hyaluronic acid (2.2, 2.1, and 1.4 × 10^3^ kDa, respectively). In this work, they also confirmed an inverse relationship between the molecular weight of hyaluronic acid and the specific cell growth rate because, since the doubling time of a cell is longer, the cells have more time to elongate the hyaluronic acid chain.

Jagadeeswara Reddy et al. [71] reported the optimization of fermentation conditions to produce hyaluronic acid by *Streptococcus nonepidemic* mutant. Using the batch mode with the following fermentation conditions: pH 7.2, 36 °C, 400 rpm, and 0.6 vvm, they achieved hyaluronic acid with a concentration of 1.84 g/L. In another study, using the same *Streptococcus* strain and the same fermentation conditions, scientists reached a higher hyaluronic acid concentration (2.34 g/L) with 2.5 × 10^3^ kDa MW using the fed-batch mode [22].

The fed-batch mode also proved to be advantageous in hyaluronic acid production by *Corynebacterium glutamicum* mutant bacteria. Cheng et al., 2016b [28] explored the hyaluronic acid production in batch mode at 28 °C, pH 7.2, 600 rpm, and 1 vvm. The hyaluronic acid production reached 8.3 g/L at 48 h, and the molecular weight of the hyaluronic acid was 1.30 × 10^3^ kDa. On the other hand, the authors performed the fermentation process using the fed-batch mode, and the hyaluronic acid concentration was enhanced to 21.6 g/L under the same fermentation conditions when produced by the same strain in the same production media concerning the process described above [29].

The culture conditions and culture media that have been explored for the microbial production of hyaluronic acid by various bacteria strains in different culture modes are shown in Table 1.

## 7. Conclusions

In conclusion, the production of hyaluronic acid using microorganisms involves a complex process that includes selecting the appropriate substrate, supplements, and culture conditions. Alternative sources, such as agricultural waste, industrial waste, and synthetic substrates can be used as substrates in the culture media, but the most appropriate substrate will depend on the microorganisms used and the scale of production. Supplementation of the culture medium with various nutrients and growth factors can also affect the rate of hyaluronic acid production. The optimal culture conditions will depend on the microorganisms used and include factors such as temperature, pH, aeration, agitation, and substrate concentration. Fermenter configuration is also important, and the design and configuration of the fermenter will depend on the scale of production and the microorganisms used. The establishment of the mode of culture is also crucial and because each mode has its own advantages and disadvantages, researchers often use a combination of different modes of culture to find the conditions that yield the highest hyaluronic acid production.

Fermentation is a widely used process to produce hyaluronic acid since it allows the optimization of yield and molecular weight due to the possibility of controlling all its main stages. Nonetheless, there are some challenges to be overcome in the microbial production of hyaluronic acid such as the limited production of hyaluronic acid due to the high viscosity of produced hyaluronic acid, competition for the same precursors between microorganism growth and HA production, accumulation of by-products in the production media, and consequently, the inhibition of hyaluronic acid production [5,7,9].

In a single process of hyaluronic acid microbial production, hyaluronic acid with different molecular weights can be obtained. Obtaining hyaluronic acid of low dispersity has been a motivation for scientists to continue to explore and optimize the process [5].

As stated above, the microorganism used most often in industrial hyaluronic acid production is *Streptococcus zooepidemicus*, although it is a pathogen. Thereby, the demand for safe hyaluronic acid producer microorganisms has also been a challenge. Investigators have used metabolic engineering to obtain recombinant microorganisms capable of producing hyaluronic acid with desired concentration and MW, such as *Bacillus subtilis* and *Corynebacterium glutamicum* [19,23,27,28,29,61].

Finally, at the industrial scale, is important to maintain the low cost of hyaluronic acid production. To address this, researchers have studied alternative culture media, such as molasses, cheese whey, and tuna peptone, among others [35,36,39,45]. Furthermore, the most significant cost in hyaluronic acid production is the downstream process, so different downstream strategies have been explored to achieve hyaluronic acid with high purity to be applied in medical fields [16].

## Figures and Tables

**Figure 1 molecules-28-02084-f001:**
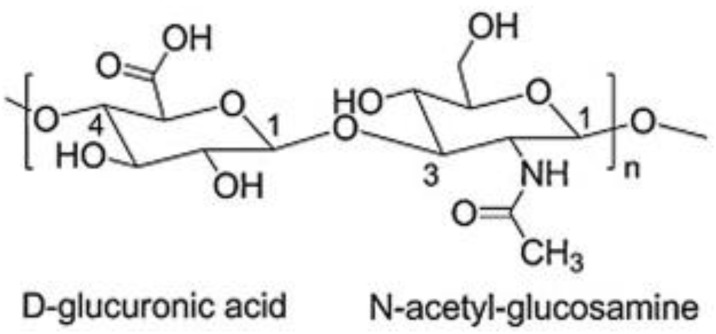
Chemical structure of HA adapted from [1].

**Table 1 molecules-28-02084-t001:** Different fermentation conditions and culture modes for producing microbial HA by various microorganisms in different culture media.

Microorganism	Culture Media	Fermentation Conditions	Culture Mode	[HA]/MW	References
*Streptococcus. Zooepidemicus* ATCC 35246	Cheese whey protein Glucose Yeast extract Tryptone	pH 6.7 T 37.00 °C 500 rpm 1.0 vvm	Batch mode 5 L bioreactor	4.0 g/L -12 h 3.71 × 10^3^ kDa 0.87 g/L h	[36]
*Streptococcus zooepidemicus* ATCC 35246	Molasses Sheep Wool Peptone	pH 8 T 37.00 °C 200 rpm 1.0 vvm	250 mL flasks	3.54 g/L-48 h	[38]
*Streptococcus zooepidemicus* ATCC 35246	Glucose Yeast extract Peptone from *Scyliorhinus Canicula* visceral treatment	pH 6.7 T 37 °C 500 rpm 0 vvm	Fed-batch mode 2 L bioreactor	2.53 g/L-18 h 2.11 × 10^3^ kDa	[42]
*Streptococcus zooepidemicus* NJUST01	Starch (5%) Glucose Peptone Yeast extract	T 37 °C 220 rpm	Batch mode 500 mL flasks	6.7 g/L-36 h	[37]
*Streptococcus sp.* ID9102 mutant strain KCTC 1139BP	Glucose Yeast extract Casein Peptone K_2_HPO_4_ MgCl_2_ Glutamine Glutamate Oxalic acid	pH 7 T 36 °C 400 rpm 0.5 vvm	Batch mode 75 L jar fermenter	6.94 g/L-24 h 5.9 × 10^3^ kDa	[34]
*Streptococcus zooepidemicus* WSH-24	Sucrose Yeast extract 3% PFC as an oxygen vector	pH 7 T 37 °C 200 rpm 0.5 vvm	Batch mode 7 L bioreactor	6.6 g/L-20 h	[49]
*Streptococcus zooepidemicus* WSH-24	Sucrose Yeast extract	T 37 °C 200 rpm aeration 2 L/min alkaline-stress fermentation	Batch mode 7 L bioreactor	6.5 g/L-16 h	[58]
*Streptococcus zooepidemicus* WSH-24	Sucrose Yeast extract	pH 7 T 37 °C 200 rpm 0.5 vvm	fed-batch mode (0–8 h) batch mode (8–20 h) 7 L bioreactor	6.6 g/L-20 h 0.023 g HA/g cell/h	[68]
*Streptococcus zooepidemicus* ATCC 39920	Glucose Yeast extract BHI Ascorbic Acid	pH 7 T 37.00 °C 200 rpm 1 vvm dilution rate 0.066 h^−1^	Chemostat mode 2,4 L Bioengineering reactor	2.6 × 10^3^ kDa	[70]
*Streptococcus thermophilus* NCIM 2904	Whey Permeate Whey Protein hydrolysate	pH 8 T 37 °C 150 rpm	Batch mode 250 mL flasks	0.34 g/L-20 h 9.22–9.46 kDa	[72]
*Bacillus subtilis* WmB	Sucrose	T 32 °C	5.0 L fermenter	3.65 g/L-54 h 3.92 × 10^2^ kDa	[61]
*Corynebacterium glutamicum/*Δ*ldh*-AB	Corn Syrup Powder Glucose IPTG (isopropyl-β-D-thiogalactoside)	pH 7.2 T 28 °C 600 rpm 1 vvm	Fed-batch mode 5.0 L fermenter	21.6 g/L-48 h 1.28 × 10^3^ kDa	[29]

[HA]/MW—concentration of hyaluronic acid (HA)/molecular weight of HA.

## Data Availability

Not applicable.

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
