# Peer review of "Microbial Hyaluronic Acid Production: A Review"

_molecules, 2023, doi:10.3390/molecules28052084_

Round 1

Reviewer 1 Report

Dear authors,

the topic of your manuscript is very interesting and the manuscript is well written. The introduction provides enough information, and the whole text is well organized, scientifically sound and supported by appropriate references. It gives an insight into microbial hyaluronic acid production, the advantages or disadvantages of certain microorganisms, fermentation conditions, culture media and supplements used in mentioned process, previous studies and new studies that could be conducted in order to investigate optimal fermentation conditions in laboratory or industrial scale.

The main question in this research is the mycrobial production of hyaluronic acid, appropriate microorganisms, fermentation conditions and optimal parameters. Hyaluronic acid has been in the centre of many studies due to its properties that make it applicable in many industries. This research gives an overview of previous studies that can help researches to set up their future experiments, and producers to select appropriate microorganism and cultivation media for fermentation process. Table 1 gives concise and very valuable information about mentioned parameters (microorganisms, culture media, fermentation conditions) supported with references. Maybe there is a way to technically organize the table to be easier to read.

Otherwise, it is my opinion to accept this manuscript in present form.

Author Response

Answer to referee’s comments and queries

Detailed responses to Reviewer 1

Reviewer´s comment: The topic of your manuscript is very interesting and the manuscript is well written. The introduction provides enough information, and the whole text is well organized, scientifically sound and supported by appropriate references. It gives an insight into microbial hyaluronic acid production, the advantages or disadvantages of certain microorganisms, fermentation conditions, culture media and supplements used in mentioned process, previous studies and new studies that could be conducted in order to investigate optimal fermentation conditions in laboratory or industrial scale.

The main question in this research is the mycrobial production of hyaluronic acid, appropriate microorganisms, fermentation conditions and optimal parameters. Hyaluronic acid has been in the centre of many studies due to its properties that make it applicable in many industries. This research gives an overview of previous studies that can help researches to set up their future experiments, and producers to select appropriate microorganism and cultivation media for fermentation process. Table 1 gives concise and very valuable information about mentioned parameters (microorganisms, culture media, fermentation conditions) supported with references. Maybe there is a way to technically organize the table to be easier to read.

Otherwise, it is my opinion to accept this manuscript in present form.

Our reply: We greatly appreciate the comments of the reviewer, who praises our work. In fact, we tried to do our best to give to the scientific community a contribution of our work in the area, and receiving these comments makes us very happy. Thanks for kind words.

Sincerely,

Ana Isabel Ramos Novo Amorim de Barros

Reviewer 2 Report

The review article summarize the very actual topic of microbial HA production and factors which affect the yield of production and Mw of HA. However, the manuscript is not written very well. The strcuture of article is logical however some sentences are not in approiate  sections e.g. lines 154-162 are about carbon source and should be rather in section about supplementation of culture media. The abbreviations are not used in consistent manner e.g. line 213, 231,257...

The manuscript should be written in more condensated form. Several information are repeated in different sections and even some sentences are repeated in article e.g. same sentence appears in 96-97, 119-120 and 138-140.

Sections 5 and 6 are better and more clear than others, the summary tables should be also added into previous sections. 

Author Response

Answer to referee’s comments and queries

Detailed responses to Reviewer 2

Reviewer´s comment: The review article summarize the very actual topic of microbial HA production and factors which affect the yield of production and Mw of HA. However, the manuscript is not written very well. The structure of article is logical however some sentences are not in appropriate sections e.g. lines 154-162 are about carbon source and should be rather in section about supplementation of culture media. The abbreviations are not used in consistent manner e.g. line 213, 231,257...

The manuscript should be written in more condensed form. Several information are repeated in different sections and even some sentences are repeated in article e.g. same sentence appears in 96-97, 119-120 and 138-140.

Sections 5 and 6 are better and more clear than others, the summary tables should be also added into previous sections. 

Our reply: We appreciate the reviewer’s comments and insightful considerations. In fact a several pertinent comments are made and were all considered in the review of the article.

Sincerely,

Ana Isabel Ramos Novo Amorim de Barros

Reviewer 3 Report

Each microorganism has its own advantages and disadvantages in terms of production performance and profitability, the selection of the appropriate microorganism depends on the specific requirements of the final application of that HA, so it would be helpful to add a paragraph on profitability and possible application of the HA produced by each  microorganism.

Author Response

Answer to referee’s comments and queries

Detailed responses to Reviewer 3

Reviewer´s comment: Each microorganism has its own advantages and disadvantages in terms of production performance and profitability, the selection of the appropriate microorganism depends on the specific requirements of the final application of that HA, so it would be helpful to add a paragraph on profitability and possible application of the HA produced by each microorganism.

Our reply: We appreciate the reviewer’s comments and insightful considerations. The information about the profitability and possible application of the HA produced by each microorganism has been introduced in the manuscript.

Sincerely,

Ana Isabel Ramos Novo Amorim de Barros

Round 2

Reviewer 2 Report

The quality of manuscript was improved. However, minor revisions are still needed. The formal english language should be used in whole article (please do not use abbreviated forms as wasn't, isn't etc. The use of abbreviations (e.g. HA, Mw) should be consistent and list of abbreviations should be added. 

Author Response

Dear Editor of Molecules,

In reply to the review performed on the paper entitled “Microbial hyaluronic acid production: A Review”, we would like to acknowledge the valuable comments performed by the editor that kindly accepted to revise our manuscript. We would like to confirm that we have addressed most issues and answered the questions made by reviewers. We hope the answers below and modifications that have been done in the manuscript are clear and concise enough as required by the reviewers to enable the publication of the manuscript in Molecules.

Answer to referee’s comments and queries

Detailed responses to Reviewer 2

Reviewer´s comment: The quality of manuscript was improved. However, minor revisions are still needed. The formal english language should be used in whole article (please do not use abbreviated forms as wasn't, isn't etc. The use of abbreviations (e.g. HA, Mw) should be consistent and list of abbreviations should be added. 

Our reply: We appreciate the reviewer’s comments and insightful considerations. In fact we agree that the abbreviations make the article more difficult to read, so we decided to remove most of them. Also, the grammatical questions were corrected in the manuscript.

Sincerely,

Ana Isabel Ramos Novo Amorim de Barros
